

# Phylogenetic analysis of *Microdochium* spp. associated with turfgrass and their pathogenicity in cereals

Filiz Ünal

Department of Plant Protection, Faculty of Agriculture, Eskişehir Osmangazi University, Eskişehir, Odunpazarı, Türkiye

## ABSTRACT

Turfgrass is frequently used today in the arrangement and aesthetic beautification of grounds in parks, gardens, median strips, recreation and sports areas. In this study, surveys were conduct in turfgrass areas in the three provinces of Türkiye. As a result of isolations from the collected samples, 44 *Microdochium* isolates obtained belonging to five different species including *M. bolleyi, M. majus, M. nivale, M. paspali* and *M. sorghi* which have different virulences. Identification of the isolates were performed by rDNA-ITS sequence analyzes. According to the pathogenicity tests results, the most virulent species was *M. nivale* M62 with a disease severity value of 91.93%. This was followed by *M. bolleyi* M1584 and *M. majus* M63, with disease severity values of 91.12% and 91.08%, respectively. The virulence of *M. bolleyi* isolates varied among the species. Only 13 of the 31 *M. bolleyi* species were found to be virulent in turfgrass, *M. paspali* was less virulent than the others in *Poa pratensis*. The most virulent isolate of each *Microdochium* species was tested on four different cereal varieties. *M. sorghi* and *M. paspali* had low virulence values in barley and oat than the other *Microdochium* species, while the other three species showed high virulence in turfgrass, wheat and barley, other than oat. In the phylogenetic neighboor-joining tree belonging to 44 *Microdochium* isolates clearly demonstrated that the isolates were grouped into five distinct clusters. *M. nivale* and *M. majus* were considered genetically close isolates.

## INTRODUCTION

Turfgrass areas, whose cultivation area increases day by day, are the center of the visual attention of people due to both their social and aesthetic appearance. In recent years, the interest in golf has led to an increase in grass areas requiring intensive maintenance. The *Microdochium* genus includes many species that are highly pathogenic, weakly pathogenic, endophytes, and saprophytes on cereals and grasses (*Kirk & Deacon, 1987*; *Hernández-Restrepo, Groenewald & Crous, 2016*; *Demirci & Dane, 2003*; *Hong et al., 2008*; *Lenc et al., 2014*; *Li, Li & Qi, 2019*; *Gagkaeva et al., 2020*; *Huang et al., 2020*). Pathogenic *Microdochium* spp. cause severe damage, especially on wheat and barley. The fungi that cause the disease called "snow mold" or "*Microdochium* patch" on wheat, especially in cool climate conditions are *M. nivale* and *M. majus*. These fungi cause economic losses in wheat

Corresponding author
Filiz Ünal, filiz.unal@ogu.edu.tr

and barley (*Tronsmo et al., 2001*; *Abdallah-Nekache et al., 2019*; *Mao et al., 2023*). These two agents also cause head blight, which is frequently observed in wheat (*Kammoun et al., 2009*; *Pancaldi et al., 2010*; *Hayashi et al., 2014*). In a study conducted in Morocco, it was reported that wheat grains were highly contaminated with *M. majus* (*Saoudi et al., 2019*). Studies conducted in various regions of the world have shown that *M. bolleyi* causes crown and root rots in grasses (*Braun, 1995*; *Hong et al., 2008*). Studies are reporting that *M. bolleyi* causes root and crown rot in wheat and barley as well as grass, however studies report that it is not a pathogen (*Hannukkala & Koponen, 1987*; *Hemens, Steiner & Schnbeck, 1992*; *Lenc et al., 2014*; *Li, Li & Qi, 2019*). For this reason, there are conflicting results regarding the hosts in which *M. bolleyi* causes disease. Another *Microdochium* species, *M. sorghi*, causes zonate leaf spots and rot on *Sorghum* spp. and other *Poaceae* (*Von Arx, 1987*; *Braun, 1995*), while *M. paspali* causes seaside paspalum disease on *Paspalum vaginatum* (*Zhang et al., 2015*). It has been reported that *M. nivale* causes basal rot, crown rot, and dryland root rot in dryland wheat fields in Türkiye, and also causes color change in seedlings or necrotic lesions in internode tissues in winter wheat (*Demirci & Dane, 2003*; *Tunali et al., 2008*).

The rDNA region of microorganisms has been used frequently and successfully in molecular diagnosis, taxonomy studies, and genetic diversity studies since the early 1980s. One of the reasons for this is the high copy number in the genome for DNA required for successful amplification, and also because the rDNA region consists of highly conserved regions that enable the design of general primers such as the ITS (*White et al., 1990*). This region is the most comprehensively sequenced and studied region in fungi (*Schoch et al., 2009*), which has enabled the creation of an enormous data pool (*O'Brien et al., 2005*). In addition to rDNA sequences such as 18S or 28S (*Hernández-Restrepo, Groenewald & Crous, 2016*; *Baghela & Singh, 2017*; *Hong et al., 2008*; *Li, Li & Qi, 2019*; *Gao et al., 2022*), β tubulin (*Myllys, Stenroos & Thell, 2002*), RNA-polymerase binding proteins 1, 2 (*Schoch et al., 2009*), and elongation factor-1α (*Schoch et al., 2009*) have been used frequently in fungi. The regions are also used successfully in molecular phylogeny. In recent years, genetic differences between species within the *Microdochium* genus have been revealed using different gene regions (*Hernández-Restrepo, Groenewald & Crous, 2016*; *Zhang et al., 2015*; *Crous et al., 2018*, *2019*; *Marin-Felix et al., 2017*; *Gao et al., 2022*).

*Microdochium* species and their metabolites have also been reported to be used in the treatment of some diseases in plants and humans. For example, bioactive compounds of *Microdochium* species can be used successfully against *Verticillium dahliae*, which causes plant vascular wilt (*Berg et al., 2005*). Cyclosporine A, a metabolite used to control animal and human diseases, has been isolated from *M. nivale* (*Bhosale et al., 2011*) and *M. phragmitis* extracts be cytotoxic in tumor cell lines in humans (*Santiago et al., 2012*). Therefore, the possibilities of using *Microdochium* species in biotechnology should be investigated in detail in the future.

This study aimed to identify *Microdochium* species isolated from turfgrass roots, to determine the pathogenicity of the obtained species in turfgrass, wheat, barley, and oats, and phylogenetic analysis of the DNA sequences of the isolates.

## MATERIALS AND METHODS

### Sample collection and isolations

Surveys were conducted in turfgrass areas of Eskişehir, Ankara, and Kocaeli provinces of Türkiye, and turfgrass samples were collected. In the isolations from the plants, tissue pieces of the diseased root and crown were dried on sterile blotting paper after 1-min surface disinfection in 1% sodium hypochlorite. Then, it was placed on potato dextrose agar (Difco, Franklin Lakes, NJ, USA). After 3–4 days, the hyphae tips of *Microdochium*-like fungi were removed with a sterile loop and transferred to PDA, and then to water agar (Merck, Darmstadt, Germany) for single spore isolation (*Lees et al., 1995*).

### Pathogenicity tests

Before pot experiments, preliminary pathogenicity tests were performed in Petri dishes to eliminate non-pathogenic isolates. Preliminary pathogenicity assays were performed according to *Ichielevich-Auster et al. (1985)* using susceptible turfgrass (*Poa pratensis*). In preliminary pathogenicity tests, 4 mm-long mycelial disks taken from cultures of the isolate grown in PDA medium at 25 °C for 3 days were transferred to 2% water agar (WA) and allowed to develop for 2 days under the same conditions. Ten seeds were then placed at the tip of the growing edge of the developing isolates in each petri dish, after the seeds of the susceptible grass variety had been disinfected in 1% NaOCl for 1 min and dried between sterile blotting papers. Fungus-free PDA discs were used as controls. Five petri dishes were used for each isolate. After 7–8 days of incubation at 25 °C, the roots and hypocotyls of the plants were examined. As a result of the assays, those showing symptoms such as light discoloration in the hypocotyl and roots, rot, and root loss were selected for pot trials. Inoculum obtained by wrapping the fungi into rye seeds was used in pot analyses. Colonies growing on potato dextrose agar medium were cut into 3 mm squares and transferred into flasks containing a twice-autoclaved mixture (50:50, by weight) of rye kernels. Flasks were incubated for 2 weeks and shaken intermittently to reduce clumping. The inoculum obtained after incubation was dried on sterile paper and chopped in a blender (*Smiley, 2019*). Then, this inoculum was mixed at 5% (w/w) into the prepared mixture (soil, sand, and manure) in a 2:1:1 ratio (sterilized twice at 121 °C for 45 min, 1 day apart). Each application was carried out in three repetitions with square pots of 12 cm × 12 cm (*Smiley, 2019*). No inoculum was added to the control pots. The pots were covered with nylon bags and left to incubate for 3 days. After incubation was completed, thirty *P. pratensis* seeds were planted in each pot and covered with soil. After planting, each pot was watered with approximately 20 ml of sterile water. Experiments were carried out in three replicates. After 4 weeks, grass plants were examined and evaluated using the 0–3 scale: 0 = no symptoms; 1 = <30% of the root color change (mild symptoms); 2 = 30–65% of the root color change (moderate symptoms); 3 = 66–100% of the root color change or the plant completely dead (severe symptoms) (*Cong et al., 2018*). The disease severity values were calculated using the disease severity value (DSV) formula. DSV(%) = $\Sigma$ (Number of plants at each rating scale × rating scale score)/(Total number of plants × maximum rating scale score) × 100 (*Townsend & Heuberger, 1943*). Finally, at the end of

the study, the reisolation of fungi from plants was carried out. As a consequence of the pathogenicity tests, isolates with disease severity values between 80–100% were considered highly virulent, isolates with disease severity values between 70–80% were considered moderately virulent, and isolates with disease severity values at 70% and below were considered as low virulent.

## Pathogenicity studies of *Microdochium* species in cereals

In the study, the most virulent isolate isolated in each *Microdochium* species was selected and their virulences were investigated in selected commercial and prevalent cultivars of turfgrass, wheat, barley, and oat varieties. Tall fescue turfgrass (*Festuca arundinacea* Schreb), Tarm-92 barley (*Hordeum vulgare* L.), Kızıltan-91 wheat (*Triticum aestivum* L.), and Seydişehir oat (*Avena sativa* L.) varieties were used in the trials. The experiments were carried out in greenhouse conditions with pot trials in three replicates. Inoculum preparation and application of inoculum to plant roots and evaluations were carried out as in pathogenicity tests. The pathogenicity tests were conducted on the cereals, disease severity values between 80–100% were considered susceptible to isolate, cereals with disease severity values between 70–80% were considered moderately susceptible to isolate, cereals with disease severity values between 70–50% were considered moderately resistant, and those below 50% were considered resistant (*Aktas & Bora, 1981*).

## Data analysis

All data were subjected to analysis of variance (ANOVA) with Statistical Version 12 (SPSS Statistics 16.0). The significance of mean differences in disease severity was determined using Tukey's test with $P = 0.05$ as a significance threshold. The data obtained in the study were analyzed according to a completely random design.

## Molecular identification and phylogenetic evaluation

In molecular studies, ITS regions of rDNA of Microdochium isolates were amplified using the universal primers ITS-1 (5′ TCC GTA GGT GAA CCT GCGG 3′) and ITS-4 (5′ TCC TCC GCT TAT TGA TATGC 3′) (*White et al., 1990*). DNAs were isolated by using a Blood and Tissue Kit (QIAGEN), according to the instructions for use. Each PCR reaction contained 12 µL Taq DNA Polymerase 2x Master Mix, 1 µL of each primer (10 µM), and 3 µL of template DNA. The PCR reaction mix was adjusted to a final volume of 25 µL with ultrapure water. PCR amplifications were carried out on the ABI Veriti thermal cycler. The following cycling protocols were used (*Hong et al., 2008*): The thermal cycler parameters were programmed for one cycle of denaturation at 95 °C for 4 min, 35 cycles of denaturation at 95 °C for 1 min, annealing at 58 °C for 1 min, extension at 72 °C for 1 min, and a final extension at 72 °C for 7 min. PCR products were analyzed by electrophoresis on a 2% agarose gel prepared with Pronosafe (Conda, Spain) DNA dye and visualized under a UV transilluminator. Sequence analyses were performed by BM Lab (Ankara, Türkiye). Then Blast analyses of isolates were made using these sequences on NCBI (http://www.ncbi.nlm.nih.gov) to find the closest matches. Sequences were edited and aligned by the CLUSTAL W method (*Thompson, Higgins & Gibson, 1994*). The phylogenetic tree was

constructed using the ITS sequences of the fungi using the neighbor-joining method by MEGA ver 7.0 software (*Kumar, Stecher & Tamura, 2016*), and the sequence distance was calculated with the Kimura 2-parameter model (*Kimura, 1980*). Bootstrap analysis was performed with 1,000 replications to determine the support for each clade.

## RESULTS

A total of 153 turfgrass samples were collected from parks, gardens, recreation areas, stadiums, and refuges in Ankara, Eskişehir, and Kocaeli provinces, where yellowing, wilting, blight, necrosis, rotting of roots, and rings and patches on the ground were observed. As a result of the isolations made from the roots and crowns of the samples, 44 fungal isolates belonging to the *Microdochium* genus were obtained. As a result of rDNA-ITS sequencing and BLAST analysis, 31 of the 44 *Microdochium* isolates obtained were identified as *M. bolleyi*, seven as *M. nivale*, three as *M. paspali*, two as *M. majus*, and one as *M. sorghi*.

As a result of preliminary pathogenicity tests performed in petri dishes using the turfgrass-sensitive variety *Poa pratensis*, which is sensitive to *Fusarium* species, 18 species of *M. bolleyi* were evaluated as non-pathogenic and eliminated because they did not show any disease symptoms in the roots and hypocotyls. The other 13 *M. bolleyi* species caused necrosis and decay on hypocotyls. All the other *Microdochium* species subjected to preliminary pathogenicity tests caused necrosis and decay on hypocotyls. The other *Microdochium* isolates caused hypocotyl necrosis and decay in the preliminary pathogenicity tests. Pathogenicity studies were conducted in pots with 26 *Microdochium* species, which were evaluated as virulent in preliminary pathogenicity tests (Table 1). As a result of the pathogenicity studies, no statistical difference was found between *M. nivale* M62, *M. bolleyi* M1584, *M. majus* M63, and *M. bolleyi* M246 isolates, which caused disease severity of 91.93%, 91.12%, 91.08% and 90.96%, respectively, and they were determined to be the most virulent isolates. The isolate with the lowest virulence value among *M. bolleyi* isolates was *M. bolleyi* M732, isolated from Ankara with 70.04% disease severity and evaluated as moderately virulent. When *M. nivale* isolates were evaluated, six isolates except *M. nivale* M171 were determined to be highly virulent, with disease severity values between 85.04% and 91.93%. Among *M. paspali* species, the most virulent isolate was *M. paspali* M874, with a disease severity of 87.66% (Figs. 1 and 2). The disease severity in the other two *M. paspali* species was calculated at 72.12% and 77.74% and was considered moderately virulent. All *M. majus* and *M. sorghi* isolates were found highly virulent (Table 1, Fig. 1). The virulence values of *Microdochium* isolates isolated from turfgrass areas varied between as moderately and highly virulent (Table 1, Figs. 1 and 2).

When pathogenic and non-pathogenic isolates are evaluated together, most *Microdochium* isolates were isolated from Kocaeli province, while the most pathogenic isolates were isolated from Eskişehir province. In addition, highly pathogenic *M. bolleyi* isolates were obtained from Kocaeli province, which is mostly coastal and has a more temperate climate than other provinces. More isolates of the *M. nivale* species were obtained in Ankara and Eskişehir, which have a continental climate, and it was determined

**Table 1 Species, diseases severity values, location and plant origin of pathogen *Microdochium* spp. isolates from turfgrass areas.**

| Species and isolate number | Disease severity (%) | Location | Plant origin |
|---|---|---|---|
| *Microdochium bolleyi* M242 | 71.98 ± 1.54 gh* | Ankara | Root and crown |
| *M. bolleyi* M246 | 90.96 ± 2.64 a | Kocaeli | Root and crown |
| *M. bolleyi* M411 | 86.62 ± 3.16 abc | Kocaeli | Root and crown |
| *M. bolleyi* M536 | 88.42 ± 1.38 ab | Kocaeli | Root and crown |
| *M. bolleyi* M560 | 79.83 ± 7.01 bcdefgh | Ankara | Root and crown |
| *M. bolleyi* M732 | 70.04 ± 6.21 h | Ankara | Root and crown |
| *M. bolleyi* M737 | 76.22 ± 3.72 efgh | Eskişehir | Root and crown |
| *M. bolleyi* M855 | 87.22 ± 2.62 abc | Kocaeli | Root and crown |
| *M. bolleyi* M856 | 73.12 ± 2.91 fgh | Ankara | Root and crown |
| *M. bolleyi* M918 | 82.54 ± 2.63 abcdef | Kocaeli | Root and crown |
| *M. bolleyi* M1413 | 74.76 ± 3.14 fgh | Ankara | Root and crown |
| *M. bolleyi* M1584 | 91.12 ± 1.30 a | Kocaeli | Root and crown |
| *M. bolleyi* M1619 | 76.46 ± 2.53 defgh | Eskişehir | Root and crown |
| *Microdochium nivale* M62 | 91.93 ± 1.72 a | Eskişehir | Root and crown |
| *M. nivale* M90 | 85.04 ± 3.99 abcde | Ankara | Root and crown |
| *M. nivale* M91 | 88.83 ± 4.13 ab | Ankara | Root and crown |
| *M. nivale* M151 | 86.42 ± 3.76 abcd | Eskişehir | Root and crown |
| *M. nivale* M152 | 80.37 ± 2.25 bcdefg | Ankara | Root and crown |
| *M. nivale* M171 | 75.21 ± 1.28 efgh | Kocaeli | Root and crown |
| *M. nivale* M191 | 89.50 ± 1.56 ab | Ankara | Root and crown |
| *Microdochium paspali* M98 | 72.12 ± 1.57 gh | Kocaeli | Root and crown |
| *M. paspali* M867 | 77.74 ± 3.57 cdefgh | Ankara | Root and crown |
| *M. paspali* M874 | 87.66 ± 1.37 abc | Eskişehir | Root and crown |
| *Microdochium majus* M63 | 91.08 ± 2.48 a | Eskişehir | Root and crown |
| *M. majus* 473 | 87.58 ± 2.64 abc | Ankara | Root and crown |
| *Microdochium sorghi* M241 | 83.08 ± 3.41 abcdef | Kocaeli | Root and crown |

**Note:**
* Different lowercase letters in column indicate significant differences at $P < 0.001$ probability level according to Tukey's Test

that *M. nivale* species obtained from these two provinces were more virulent in turfgrass (Table 1).

By selecting the most virulent isolates of five species isolated from turfgrasses, their virulences were investigated in tall fescue turfgrass (*Festuca arundinacea* Schreb), Tarm-92 barley (*Hordeum vulgare* L.), Kızıltan-91 wheat (*Triticum aestivum* L.), and Seydişehir oat (*Avena sativa* L.) varieties, which are widely used in Türkiye (Table 2).

*F. arundinacea* was found to be susceptible to all five *Microdochium* species, showing disease severity values between 79.46% and 92.96%. The most virulent species in *F. arundinacea* was *M. bolleyi*, which showed a disease severity value of 92.96%. This was followed by *M. majus* with a value of 92.18%. *M. paspali* was found to cause less severe disease (79.46%) in *F. arundinacea* compared to the other four *Microdochium* species. *T. aestivum* was found to be susceptible to all *Microdochium* species, showing disease

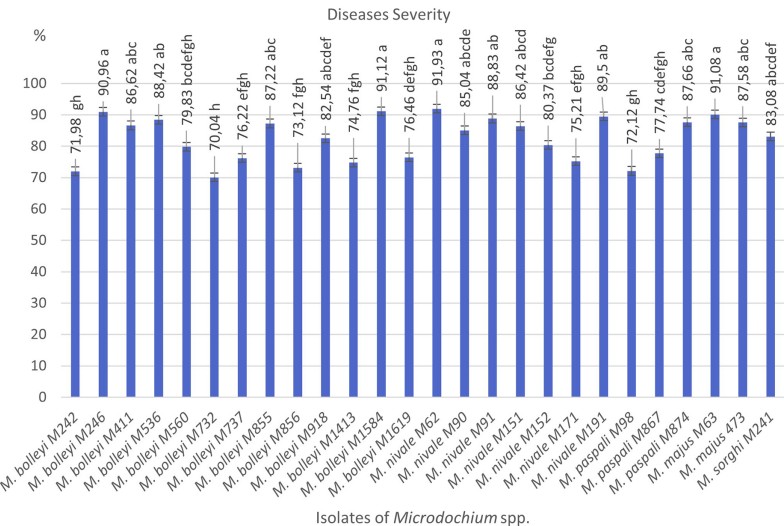

**Figure 1 Diseases severity values of isolates of pathogen *Microdochium* spp. from turfgrass areas.**

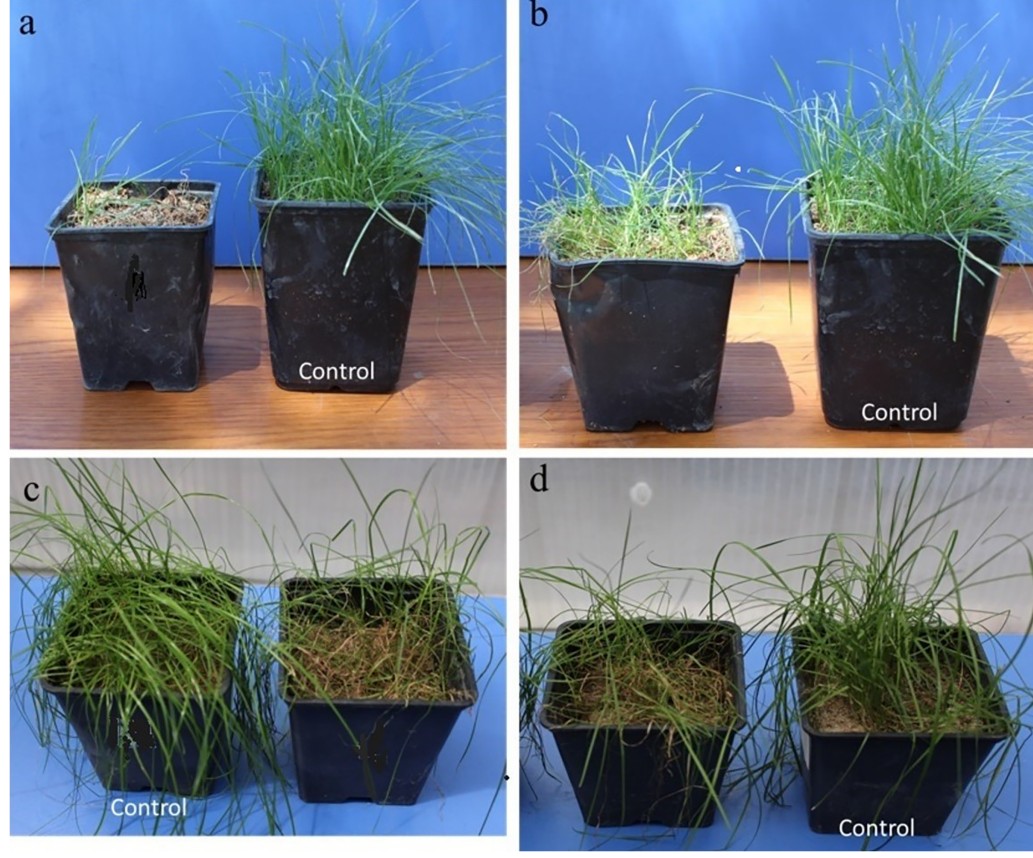

**Figure 2 Images from pathogenicity studies: (A) *Microdochium nivale* M62, (B) *M. bolleyi* M560, (C) *M. bolleyi* M536, (D) *M. paspali* M874.**

**Table 2 Diseases severity values of five *Microdochium* species on turfgrass, wheat, barley and oat.**

**Disease severity values (%)**

| *Microdochium* spp. | *Festuca arundinacea** | *Triticum aestivum*** | *Hordeum vulgare** | *Avena sativa** | Mean***** |
|---|---|---|---|---|---|
| *M. bolleyi**** | 92.96 ± 2.06 a A | 87.66 ± 1.37 A | 92.12 ± 1.57 a A | 76.33 ± 3.57 a B | 87.27 ± 7.20 A |
| *M. nivale**** | 87.12 ± 4.28 abc AB | 91.17 ± 4.35 A | 86.48 ± 5.34a AB | 75.91 ± 4.10 a B | 85.17 ± 705 A |
| *M. sorghi**** | 84.37 ± 3.88 bc AB | 86.83 ± 3.49 A | 74.67 ± 4.07bc BC | 68.58 ± 4.79ab C | 78.61 ± 8.44 AB |
| *M. paspali**** | 79.46 ± 1.71 c A | 83.29 ± 3.09 A | 69.75 ± 2.88 c B | 63.29 ± 3.09 b B | 73.95 ± 857 B |
| *M. majus**** | 92.18 ± 1.98 ab A | 91.52 ± 4.00 A | 82.87 ± 6.36 ab AB | 77.66 ± 1.37 a B | 86.06 ± 7.19 A |
| **Mean**** | 87.22 ± 5.77 AB | 88.09 ± 4.27 A | 81.18 ± 9.12 B | 72.36 ± 6.48 C | 82.21 ± 9.06 |

Notes:
* Different lowercase letters in the columns indicate significant differences between pathogens in each plant genera at least $P < 0.01$ probability level.
** In the column the pathogens were non-significant in the plant genus at $P < 0.05$ probability level.
*** Different capital letters in the lines indicate significantly differences between the plant genera in each pathogen at least $P < 0.01$ probability level.
**** Different capital letters in last line indicate significantly differences between plant genera in the mean of pathogens at least $P < 0.01$ probability level.
***** Different capital letters in last column indicate significantly differences between pathogen genera in mean of plant genera at least $P < 0.01$ probability level according to Tukey's Test.

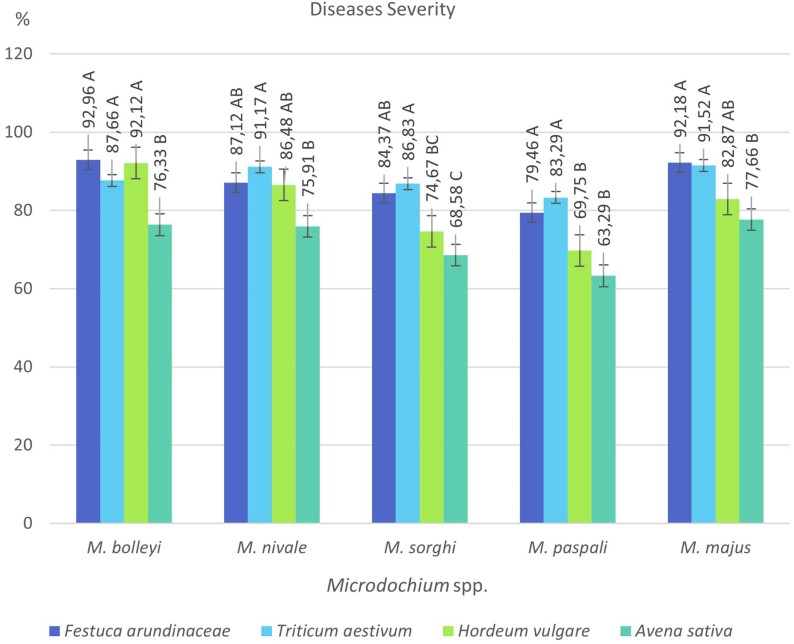

**Figure 3 Disease severity values of five *Microdochium* species on turfgrass, wheat, barley and oat.**

severity of 83.29–91.52%. The species that caused the most severe disease in *T. aestivum* was *Microdochium majus*. *Hordeum vulgare* was found to be sensitive to *M. bolleyi*, *M. majus*, and *M. nivale*, while it was found to be moderately sensitive to *M. sorghi* and *M. paspali* compared to the other three species, with disease severity values of 69.75% and 74.67%, respectively. While *Avena sativa* showed a moderately resistant reaction to *M. paspali* with a disease severity value of 63.29%, it showed a moderately sensitive reaction to the other four species (Table 2, Figs. 3 and 4).
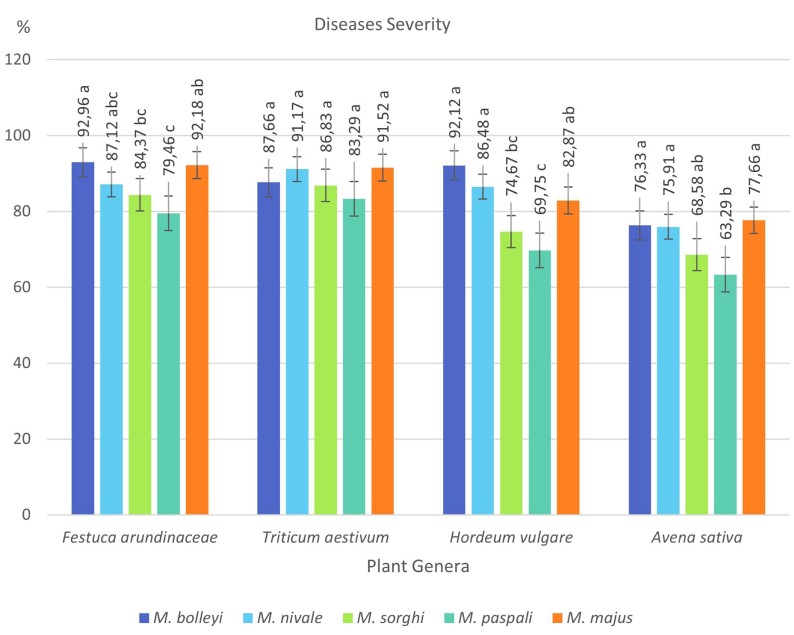

**Figure 4** Disease severity values of four plant species under *Microdochium* spp. inoculations.

A phylogenetic tree with the *Microdochium* isolates isolated from turfgrass was constructed by bootstrap neighbor-joining analysis of nucleotide sequences to evaluate genetic differences among isolates belonging to five *Microdochium* species. The Neighboor-Joining phylogenetic tree belonging to *Microdochium* isolates demonstrated that the isolates were grouped into five distinct clusters (Fig. 5). It was observed that *M. nivale* and *M. majus* species formed their own small groups within the same group on the tree. This situation showed that these two species were genetically closely related species. Isolates belonging to the other four species were grouped within themselves and formed groups in different places than other species (Fig. 5).

## DISCUSSION

The aim of this study was to identify *Microdochium* species isolated from turfgrass fields, to investigate the pathogenicity of the identified species on turfgrass, wheat, barley, and oat, and to investigate the genetic variation among the isolates. As a result of the surveys carried out in the spring, it was determined that the dominant species in Eskişehir, Ankara, and Kocaeli provinces was *M. bolleyi*. This was followed by *M. nivale*. *M. bolleyi* was generally isolated from the seaside province of Kocaeli, and the virulence of *M. bolleyi* species isolated from this province was higher than the others. This suggests that *M. bolleyi* is better adapted to regions with humid and temperate climates. *M. nivale, M. Majus*, and *M. sorghi* were more isolated from Ankara and Eskişehir, which have a continental and relatively colder climate. It is known that *M. nivale* survives even at low temperatures and causes serious infections, especially in grass and wheat (*Tronsmo et al., 2001*). In a study conducted in China, it was reported that *M. paspalum* caused leaf necrosis in the grass species *Paspalum vaginatum* on golf courses in late winter and early spring.

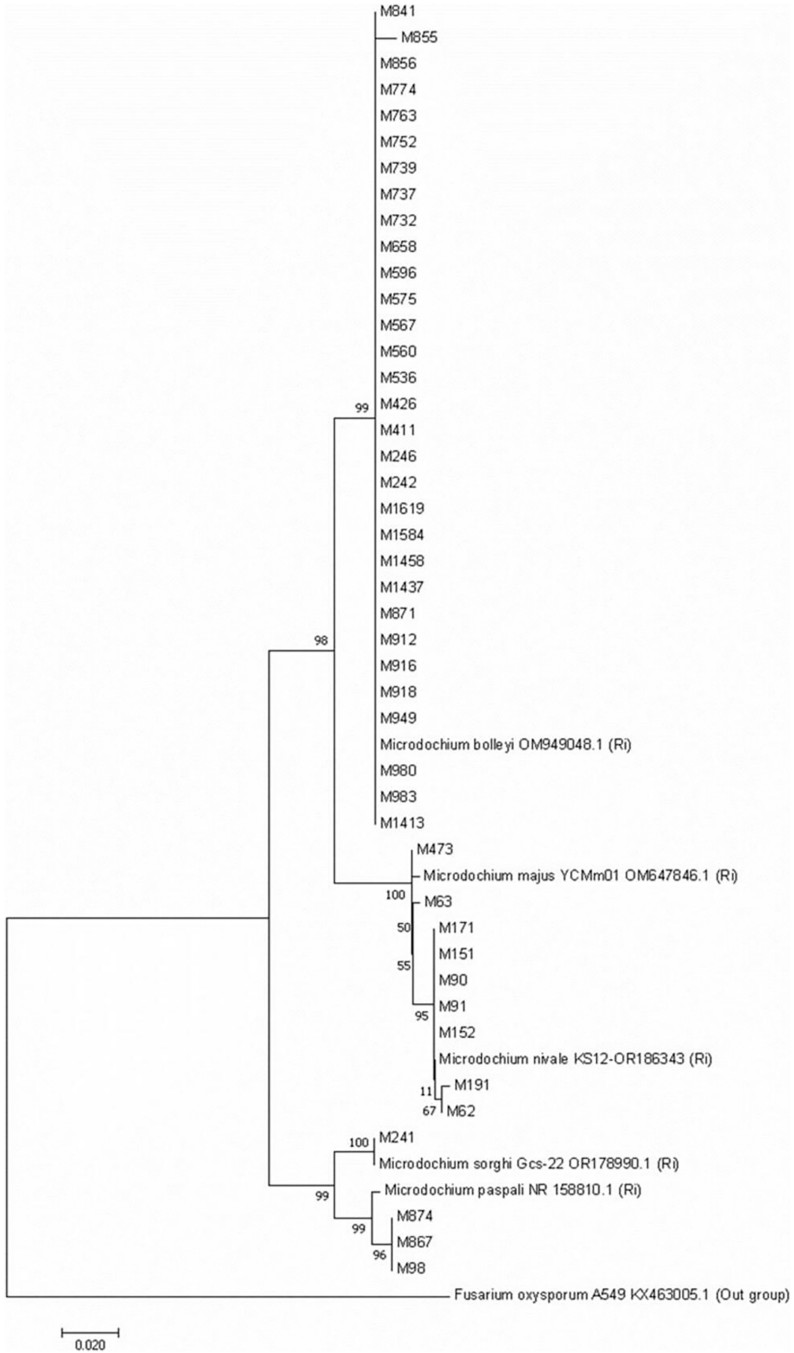

**Figure 5 Neighbour-Joining phylogenetic tree of *Microdochium* isolates obtained with 1,000 bootstrap replicates.**

In this study, while all of the *M. nivale*, *M. paspali*, *M. Majus*, and *M. sorgi* species isolated from grass areas were found to be pathogenic in turfgrass, only 13 of the 31 *M. bolleyi* species isolated were found to be pathogenic in turfgrass. Pathogenicity values of *M. bolleyi* isolates also varied.

While it was stated in some previous studies that *M. bolleyi* was a non-pathogenic, endophytic (*Mandyam, Loughin & Jumpponen, 2010*) and weak parasitic (*Kirk & Deacon, 1987*) agent, in some recent studies, it was found to be a pathogen, especially in turfgrasses (*Braun, 1995*; *Hong et al., 2008*). In a study conducted in Finland, it was reported that *M. bolleyi* is a fungus commonly found on barley and wheat roots (*Hannukkala & Koponen, 1987*). Additionally, *M. bolleyi* is reported to be isolated from diseased roots, often together with other fungi that cause disease in cereals. In the same study, it was reported that the agent also colonized other plants other than oats and cereals, although not as much as wheat and barley. *Hemens, Steiner & Schnbeck (1992)* stated that *M. bolleyi* develops within roots and coleoptiles of barley either leading to damage of the tissue or a symptomless infection. They indicated that the hyphae of the agent were seen in both, roots and coleoptiles in dead cells only. On the other hand, it was reported that *M. bolleyi* caused root rot on naked barley in China (*Li, Li & Qi, 2019*). It has even been reported to cause damping-off on wheat (*Lenc et al., 2014*). In this study, although the majority of *M. bolleyi* Türkiye isolates were determined to be non-pathogenic, pathogenic isolates were determined to be highly virulent, especially in grass, wheat, and barley. Wheat, barley, and grass in particular were found to be susceptible to virulent strains of the agent. Similarly, it has been reported that *M. bolleyi* causes basal rot and causes significant damage to turfgrasses on golf courses in Korea (*Hong et al., 2008*).

In this study, all *M. majus* and *M. nivale* isolates from turfgrass were highly pathogenic on turfgrass and the other cereals. These two species were isolated especially from terrestrial and cold areas and their disease severity values were found to be quite high. It was stated that *M. nivale* species are common plant pathogens in cool and temperate regions of the northern hemisphere (*Lees et al., 1995*; *Tronsmo et al., 2001*; *Waalwijk et al., 2003*; *Hong et al., 2008*; *Kammoun et al., 2009*). Although some studies indicated that *M. majus* is relatively less pathogenic than *M. nivale* on grasses (*Holmes, 1976*; *Hofgaard et al., 2006*), in this study, *M. majus* caused disease severity similar to *M. nivale*. *M. majus* caused brown foot rot in wheat (*Mao et al., 2023*). Both species cause significant damage to cereals (*Diamond, Cooke & Dunne, 1995*). In this study, all *M. paspali* species were found to be moderately pathogenic in turfgrass. *M. paspali* species were isolated from all three provinces, and the most virulent *M. paspali* strain was isolated from Eskişehir, Türkiye. In a study conducted in China, it was reported that *M. paspali* caused serious damage to golf courses in regions with cold climate periods (*Zhang et al., 2015*). *M. sorghi* causes zonate leaf spots and decays on *Sorghum* species and other *Poaceae* (*Von Arx, 1987*; *Braun, 1995*). In this study, while *M. sorghi* was highly virulent in turfgrass and wheat, it was found to be moderately virulent in barley and oats. In the future, studies should be conducted to reveal the virulence status of *Microdochium* species isolated from different geographical regions and climatic conditions in *Poaceae*.

*Microdochium* species have been reclassified with molecular studies conducted in recent years (*Glynn et al., 2005*; *Jewell & Hsiang, 2013*; *Hernández-Restrepo, Groenewald & Crous, 2016*). For instance, *Glynn et al. (2005)*, as a result of their analysis using the translation elongation factor 1-alpha gene (TEF1), showed that there were significant differences between the isolates previously identified as *M. nivale* and that these isolates were two

separate species, *M. nivale* and *M. majus*. In this study, with different *Microdochium* species isolated from turfgrass, a Neighbour-Joining phylogenetic tree was constructed with 1000 bootstrap replicates. The phylogenetic tree was constructed based on the internal transcribed spacer (ITS) genetic region of 44 isolates including different *Microdochium* isolates. ITS is the most commonly used in phylogenetic studies of fungal isolates, followed by EF-1α, β-tubulin and RPB2 (*Hibbett et al., 2016*). Genetic differences between species in five different *Microdochium* species were detected on the tree. The results obtained in the current study support the suggestion of *Glynn et al. (2005)*. In their studies, researchers stated that *M. nivale* and *M. majus* should be considered as two separate species. However, *Jewell & Hsiang (2013)* stated in their study that they could not distinguish between these two fungal species using ITS sequences. They reported that the multicopy nature of the ITS sequence facilitates amplification from low-quality DNA but also limits the ability to distinguish between inter- and intraspesific variations (*James et al., 2006*; *O'Donnell et al., 2015*; *Hibbett et al., 2016*). In our study, *M. bolleyi*, *M. paspali*, and *M. sorghi* isolates were grouped among themselves and settled in different parts of the tree, but *M. nivale* and *M. majus* species formed two different small groups within a single large group, showing that they were two different species close to each other. It was observed that *M. nivale* species showed minor differences within themselves.

## CONCLUSIONS

In conclusion, the results obtained from the genotypic and pathogenic differences detected between isolates from this study support the species-based differences of *Microdochium* genus members. Future investigations of the relationships between host diversity and the genetic diversity of isolates and the effects of different geographical regions and climatic conditions on genetic diversity will bring interesting results. Further studies may reveal new *Microdochium* isolates, their virulence in different hosts, and their management.

### Funding
The author received no funding for this work.

### Competing Interests
The author declares that they have no competing interests.

### Author Contributions
- Filiz Ünal conceived and designed the experiments, performed the experiments, analyzed the data, prepared figures and/or tables, authored or reviewed drafts of the article, and approved the final draft.

### Data Availability
The raw measurements are available in the Supplemental Files.

## Supplemental Information

Supplemental information for this article can be found online at http://dx.doi.org/10.7717/peerj.16837#supplemental-information.

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
