# Peer review of "Phylogenetic analysis of Microdochium spp. associated with turfgrass and their pathogenicity in cereals"

_PeerJ, doi:10.7717/peerj.16837_

## Round 0.1 · original submission · Major Revisions

Dear Authors

The manuscript cannot be accepted for publication in its current form. It needs a major revision to be reconsidered for publication. The authors are invited to revise the paper considering all the suggestions made by the reviewers. Please note that requested changes are required for publication.
With Thanks

**Language Note:** The review process has identified that the English language must be improved. PeerJ can provide language editing services - please contact us at copyediting@peerj.com for pricing (be sure to provide your manuscript number and title). Alternatively, you should make your own arrangements to improve the language quality and provide details in your response letter. – PeerJ Staff

Reviewer 1 ·

Basic reporting

The "Phylogenetic analysis of Microdochium spp. associated with turfgrass and their pathogenicity in cereals" addresses pathogenicity and phylogeny of Microdochium spp. on four hosts. Overall, the manuscript is well-written and the results are clearly presented.
However there are several typos in the manuscript. (Please see the attached file).

Experimental design

Research questions are relevant and dwell defined.
Methods have been described with sufficient information.

Validity of the findings

The results and conclusions stated appropriately.
The statical analysis have been presented in the tables, but it would be helpful if you add significance
of mean differences on the figures.

Additional comments

Please check the attached manuscript. There are several issues that should be addressed to be worthy of publication.

Annotated reviews are not available for download in order to protect the identity of reviewers who chose to remain anonymous.

·

Basic reporting

The manuscript is well written and presented excellent information. Author analyzed various species of Michrodochium spp. I strongly recommend, the manuscript can be accepted after minor changes/corrections mentioned through the manuscript.

Experimental design

The experiment follows a logic sequence to help the reader to understand what happened and what should be happen in the future studies.

Validity of the findings

The author did pathogenicity tests and also used molecular markers for further confirmation of Fungi she found in her study. So the finding of this study are clear and has meaningful replication and statistic analysis behind it.

Additional comments

Author suggested to do the correction made in word format of the manuscript.

Reviewer 3 ·

Basic reporting

The manuscript "Phylogenetic analysis of Microdochium spp. associated with turfgrass and their pathogenicity in cereals" has good and reasonable results. However, there are some items that should be considered by the authors and revised well before acceptance, as follows:

1- The whole text should be polished and edited well by a fluent English-speaking expert. Several sentences are not clear in meaning, or even have false and incorrect meaning.

2- In the introduction section, the aim of the study was not mentioned clearly. The study was identification of Microdochium species especially using molecular technique and in the next step evaluation of their potential of pathogenecity on different cereals. There is no study on genetic variation since no genetic diversity markers were not used. So, the aim of the study should be focused on taxonomy as well as pathogenecity test.

3- The figure 1 is in Turkish language. It should be corrected in English language. Also, the numbers and values in top of each bar should be arranged well. Some of them were mixed to each other. The figures 2 and 3 could be used as sample.

4- The figures 2 and 3 were designed well, but there are no statistical grouping in the top of the bars. It is needed to put the grouping with letters or put the error bars.

Experimental design

In method and materials section, there are some items that should be clarified as follows:

1- There is no map or GPS coordinates of sampling regions and locations. Please provide this information for clarification.

2- For molecular identification, at first steps, there is needed to identify the Microdochium genus and species using morphological methods. There is no information in the manuscript regarding how the genus and species were identified morphologically. The methods and identification keys should be clarified well.

3- For better clarification, it will be good that include some pictures and photos of pathogenecity test (preliminary as well as main test) and also the gel photos (molecular studies). Generally, in similar studies, the authors show the results of pathogenecity and also the molecular studies with descriptive pictures and photos.

4- In the section of preparation of the fungal inoculum for pathogenecity tests, there is no description of preparation method with reference. It is needed to describe the details of inoculum preparation with stating a reference.

5- Why the authors did not use of sterilized soil in pathogenecity tests in pot experiments? Normally, in the pathogenecity tests due to their sensitivities, the mixture which are used in pots should be sterilized.

6- In pathogenecity test, the authors used from 1-3 scale but did not describe well these scales. What is the meaning of 1 or 2 or 3 scales. It should be well clarified. Also, the authors estimated the disease severity, but the formula should be mentioned in the text.

7- For pathogenecity test on the cereals, what is the basis for selection of the specific varieties of the wheat, barley, turfgrass and oat. If they are the commercial and prevalent cultivars, or they are susceptible cultivars. It should be mentioned in the text.

8- There is no information regarding the type of experimental design used both for preliminary and main pathogenecity tests. It should be mentioned clearly in the text.

9- It is better that authors consider the statistical analysis in a separate heading and describe all the data statistical analysis as well as molecular analysis in this section.

10- In DNA extraction, why the plant and fungi extraction kit was not used. The authors used from Blood kit. Why? Since in almost 90% of papers published, the fungal DNA was extracted manually or by plant and fungal kits. If there is any special reference? If yes, please indicate.

11- There is no information regarding the primers used in this study. The sequences of used primers should be mentioned in the text.

Validity of the findings

There are some items that should be considered and clarified well in results section as follows:

1- The authors used statistical analysis such as ANOVA and means grouping. In the result section, it is needed to show the ANOVA table to show the significant difference between groups. Also, there are tables with statistical grouping of the disease severity values and also the graphs but unfortunately the similarity and dissimilarity as well as the significant differences among treatments are not described well in the result section. The values mentioned but not clarified well in statistical view. It is needed that the differences were clarified well based on the results that showed in the tables as well as graphs.

2- As mentioned before, there are some items in the graphs and photos as well as in English structures that should be revised and polished by the authors.

Additional comments

As mentioned before in the comments, There are some items that should be considered by the authors and revised.

---

## Round 0.2 · Minor Revisions

Dear Author

The manuscript still needs a very minor revision as per the comment from R3.

With Thanks

Reviewer 1 ·

Basic reporting

The revised manuscript has been improved significantly by implementing the suggestions and remarks in terms of English grammar and structure.

Experimental design

Research questions and methods have been described with sufficient information.

Validity of the findings

All data have been provided which are clear.
The results and conclusions stated properly.

·

Basic reporting

All the requested changes by reviewers were done through the manuscript and question of the reviewers were replied. So the paper can be accepted for publishing.

Experimental design

They were explained very clearly and all the questions of the reviewers were answered very carefully.

Validity of the findings

It was written very clearly.

Additional comments

Now it can be accepted for publishing.

Reviewer 3 ·

Basic reporting

The manuscript was well edited and polished.
However, again, there is the following problem that should be corrected and revised in the text.
1- I think that the authors should delete the genetic differences between isolates from the hypothesis statement in introduction section. The work concentrate on phylogeny of Microdochium genus based on molecular technique. The genetic differences only will be defined by markers such as RAPD, SSR, RFLP,..... so it should be deleted from the hypothesis and the aims of the study.

Experimental design

The experimental design was good and revised

Validity of the findings

It is o.k

---

## Round 0.3 · accepted · Accept

Dear Author

I am pleased to inform you that after the last round of revision, the manuscript can be accepted for publication. Congratulations on accepting your manuscript, and thank you for your interest in submitting your work to PeerJ.

Best Regards